# Barriers to healthcare access among reproductive age women in extremely high and very high maternal mortality countries: Multilevel mixed effect analysis

Wubshet Debebe Negash[1]*, Adina Yeshambel Belay[1], Lakew Asmare[2], Demiss Mulatu Geberu[1], Asebe Hagos[1], Melak Jejaw[1], Kaleb Assegid Demissie[1], Misganaw Guadie Tiruneh[1], Kaleab Mesfin Abera[3], Yawkal Tsega[4], Abel Endawkie[2], Nigusu Worku[1], Amare Mesfin Workie[5], Lamrot Yohannes[6], Mihret Getnet[7,8]

1 Department of Health Systems and Policy, Institute of Public Health, Collage of Medicine and Health Science, University of Gondar, Gondar, Ethiopia, 2 Department of Epidemiology and Biostatistics, School of Public Health, Collage of Medicine and Health Science, Wollo University, Dessie, Ethiopia, 3 Department of Health Systems and Policy, Institute of Public Health, Collage of Medicine and Health Science, Wollo University, Dessie, Ethiopia, 4 Department of Health Systems and Management, School of Public Health, Collage of Medicine and Health Science, Wollo University, Dessie, Ethiopia, 5 Department of Nutrition, Institute of Public Health, College of Medicine and Health Science, University of Gondar, Gondar, Ethiopia, 6 Department of Environmental and Occupational Health and Safety, Institute of Public Health, College of Medicine and Health Science, University of Gondar, Gondar, Ethiopia, 7 Department of Human Physiology, School of Medicine, College of Medicine and Health Science, University of Gondar, Gondar, Ethiopia, 8 Department of Epidemiology and Biostatistics, Institute of Public Health, College of Medicine and Health Science, University of Gondar, Gondar, Ethiopia

* wubshetdn@gmail.com

**Data Availability Statement:** The data used for this study is attached as a supplementary file (Supplementary file 1).

## Abstract

### Background

It is widely recognized that maternal deaths in low-resource countries are attributed to deprived access to maternal health services. Therefore, the aim of this study was to assess barriers to healthcare access among reproductive age women in extremely high and very high maternal mortality countries.

### Methods

A community based cross sectional surveys were conducted among 181,472 reproductive age women. Stata version 17.0 was used to analyze the data. Mixed effect binary logistic regression model was analyzed. Odds ratio along with 95% CI was generated to identify factors associated with barriers to healthcare access. A p-value less than 0.05 was declared as statistical significance.

### Results

A total of 64.3% (95% CI: 64.06, 64.54) reproductive age women faced barriers to healthcare access. Young age, no formal education, poor wealth index, no media exposure,

**Funding:** The author(s) received no specific funding for this work.

**Competing interests:** The authors declare that they have no competing interests.

**Abbreviations:** AOR, Adjusted Odds Ratio; DHS, Demographic and Health Survey; ICC, Intra Class Correlation coefficient; LLK, Log Likelihood; MOR, Median Odds Ratio; PCV, Proportionate Change in Variance; SSA, Sub-Saharan Africa; SDG, Sustainable Development Goal; UHC, Universal Health Coverage.

multiparty, no health insurance coverage, and rural residence were significantly associated with barriers to healthcare access.

## Conclusion

More than six in ten reproductive age women had barriers to healthcare access in extremely high and very high maternal mortality countries. Increasing extensive health education, minimizing financial hardship by expanding health insurance may minimize barriers to healthcare access with attention to rural resident reproductive age women.

## Background

Accessibility of healthcare has many dimensions including availability of service, the quality of service offered, geographical accessibility, and financial accessibility [1]. A person's physical and mental health, as well as their overall quality of life, may be adversely affected by barriers to accessing health services [2]. Individuals' health, as well as their physical, emotional, and social well-being, is affected by their access to healthcare [3]. Having access to comprehensive, high-quality care is essential to good health, preventing and treating diseases, reducing disability risk, and achieving health equity [3].

The World Health Organization recommends every nations to be responsible for ensuring access to healthcare services as a fundamental human right [1]. To achieve Sustainable Development Goal (SDG-3) targets 3.7 and 3.8, national strategies and programs must integrate into reproductive healthcare including information and education about family planning [4, 5]. As part of SDG target 3.1, it is also aimed at reducing maternal mortality to less than 70 deaths for every 100,000 live births by 2030 [6]. It is clear from the evidence that many women face hurdles in their quest to access healthcare, resulting in poor health outcomes such as miscarriages, unsafe abortions, stillbirths and maternal death [7].

It is estimated that eight million people die annually as a result of treatable medical problems, and 400 million people do not have access to healthcare services [8]. A majority of the African population lives more than a hundred miles from modern health care facilities, primarily due to geographical distance and financial constraints [9, 10]. In low resource setting countries, due to weak health system and delivery, maternal mortality remains the main health problem [11, 12]. There were 295,000 maternal deaths worldwide in 2017, and 94% occurred in low-income countries [13], with two thirds occurring in sub-Saharan Africa. In these settings, low access to healthcare is a major contributing factor to maternal deaths [14].

Barriers to healthcare access generate a situation where health needs of the people are not fulfill or failure to have healthcare, leading financial problem and excessive admission [3]. As a solution for the above problems, many countries in sub-Saharan Africa have universal health coverage as their national health policies for the purpose of ensuring healthy lives for people of all ages [15]. However, according to the 2023 UNICEF report, countries of south Sudan and Chad had extremely high (>1000) maternal mortality countries [16]. Nigeria, Central African Republic, Guinea-Bissau, Liberia, Somalia, Afghanistan, Lesotho, Guinea, Democratic Republic of the Congo, Kenya, and Benin were reported as very high (500–999) maternal mortality countries [16].

One of the underlying cause of high maternal mortality in low income countries is barriers to accessing maternal health services [17–19]. Assessment of accessibility to healthcare issues

in high maternal mortality countries can guide decision makers towards taking necessary measures for appropriate and equitable healthcare [20]. Thus, to get holistic picture about barriers to healthcare for women in sub-Saharan Africa need to be evaluated empirically for up to date intervention by respective country governments and other partners working on maternal health. Therefore, the aim of this study was to assess magnitude of barriers to healthcare access and associated factors among reproductive age women in extremely high and very high maternal mortality countries.

## Methods

### Study design and settings

A community based assessment was conducted between January 2014 to December 2022 among reproductive age women. The survey was conducted in Afghanistan, Benin, Chad, Central African Republic, Democratic Republic of Congo, Kenya, Guinea, Lesotho, Liberia and Nigeria. In the year 2023, UNICEF reported extremely high maternal mortality (>1000) at South Sudan, Chad and Nigeria. There is also very high maternal mortality (500–999) at Afghanistan, Benin, Somalia, Democratic Republic of Congo, Kenya, Guinea, Lesotho, and Liberia. Somalia, South Sudan and Guinea Bissau had no Demographic Health Survey (DHS) data. Therefore, these countries were not included in the analysis of the current study [16]. Despite there is DHS for Central African Republic, we excluded the country because the data was collected 30 years back which is out dated (1994) [16]. It is possible to access the data from the official database of the DHS program, https://dhsprogram.com after authorization has been granted via online request by explaining the purpose of our study. Women's individual record (IR file) data set and extracted all the included factors. A nationally representative data of DHS is conducted every five year interval across low and middle income countries [21]. The data mainly includes the reproductive health issues of the mothers [21]. Basically, two stage stratified sampling technique was used. A total of weighted 181, 472 reproductive age women were included for the analysis.

### Study variables

Barriers to healthcare access was the dependent variable for this study. The variable was assessed by four questions about challenges in accessing healthcare as follows: whether or not the woman faced difficulties in gaining money, distance to get the health facility, permissions for treatment and companionship. If the women faced at least one of the listed problems, she was considered as barriers to healthcare access and recoded as 1, whereas, those women who did not reported challenges in either money or distance or permission or companionship were considered as not faced barriers to healthcare access and codded as 0 [1, 8].

**Independent variables.** Different independent variables were considered in this study to determine factors associated with barriers to healthcare access (Table 1).

### Modeling approaches

Multilevel logistic regression model was used to identify both individual and community level factors of barriers to healthcare access. STATA version 17 software was used for analysis. During each analysis, the data was weighted as v005/1,000,000 for ensuring the DHS sample representative and to obtain reliable estimations. Bivariable analysis that calculated the proportion of barriers to healthcare access with each independent variables was employed. Variables that resulted p-value less than 0.05 in the bivariable analysis were considered for multivariable regression. Finally, multilevel logistic regression analysis comprising both fixed

**Table 1. List of variables for the assessment of barriers to healthcare access among reproductive age women in extremely high and very high maternal mortality countries.**

| Variables | Description |
|---|---|
| Age of women | 15–24, 25–34, 35 & above |
| Type of place of resident | Rural, Urban |
| Educational status | No formal education, Primary education, Secondary & higher education |
| Wealth index | In the DHS wealth index is classified into five categories using principal component analysis as poorest, poorer, middle, richer and richest |
| Current marital status | Married, unmarried |
| Media exposure | Those respondents who watched television or listen radio or read newspaper/magazine for at least once a week were considered as having media exposure otherwise. On the other hand those women who did not exposed for at least one of the aforementioned medias were considered as no media exposure |
| Parity | Nulli parity, multipara (1–3), grand multipara (4 or more) |
| Covered by health insurance | In the DHS, coverage of health expenditure by health insurance is categorized as Yes or no responses. |
| Household head | Male, female |
| Community level factors | The community level education, media exposure and poverty were formed by aggregating their corresponding individual level variables at cluster level. Finally, these variables were categorized by using median as high and low, because these were not normally distributed. The community level education was generated by the proportion of households in the educated categories. Categorized as low if the proportion of women were educated below 50% and high if the proportion is ≥50%. [22]. Community level Poverty was aggregated by the proportion of households in the poorest and poorer quintile. Grouped as low if the proportion from a given community is <50% and high if the proportion is ≥50%. Community level media exposure was generated by the proportion of media exposure. Categorized as low if the proportion from a given community is < 50% and high if the proportion is ≥50% [22]. |

and random effect was conducted. Fixed effect of the model was presented as adjusted odds ratio. On the other hand random effects were expressed with intra-class correlation coefficient (ICC) [23]. Four models were analyzed: null model expressing the variations in barriers to healthcare access. Model **I**: adjusted for individual level variables. Model **II**: Adjusted for community level variables, Model **III**: adjusted for both individual and community level factors with the outcome variables [23, 24]. Deviance or Log likelihood was used as model comparison.

**Patient and public involvement statement.** Reproductive age women were included in this study by providing valuable information. Nevertheless, they have never been involved in the study design, protocol, data collection tools, and reporting disseminating the finding.

**Ethics approval and informed consent.** Demographic Health Survey website link (www.measuredhs.com) was used to obtain ethical approval and permission to access the data. According to United States Department of Health and Human Services requirements for the protection of human subjects, all methods were approved by ICF International and an Institutional Review Board (IRB. Ref/194622/23) in Ethiopia. In accordance with ethical standards, the consent manuscript was reviewed by the Institutional Review Board of the Demographic and Health Surveys (DHS) program data archivists following submission to the DHS program/ICF International. All subjects and/or their legal guardians of minors under the age of 16 provided informed consent. A third party was not given access to the data set. The study is non experimental. Further explanation of how the DHS uses data and its ethical standards can be found at: http://goo.gl/ny8T6X.

## Results

### Individual level factors of barriers to barriers to healthcare access

A total weighted 181, 472 reproductive age women were included for this study. Most of the study participants (36.66%) were youths. Nearly half (40.64%) of the mothers had no formal education. Almost all (97.89%) of the respondents healthcare did not covered by health insurance (Table 2).

Majority (62.03%) of the study participants lived in rural area. Nearly half (47.13%) of the reproductive age women were from a population with low proportion of community level education (Table 3).

### Random effect and model fitness

Based on the intra class correlation coefficient (ICC) in the null model, 18% of the overall variability of barriers to healthcare access can be attributed to cluster variability. The median odds ratio for the barriers to healthcare access in the null model was 2.31. This indicates that, choosing an individual at random from two clusters, reproductive-aged women living in clusters with high barriers to healthcare had 2.31 times higher odds of having barriers to healthcare access compared to reproductive aged women from the lower risk to barriers to health care access cluster. Similarly, the proportional change variance (PCV) increased from 11.05% in model I to 24.95% in model III, which indicates the final model (model III) best describes the barriers to healthcare access variability. The model fitness was assessed using deviance

**Table 2. Individual level factors of barriers to healthcare access in extremely high and very high maternal mortality countries.**

| Variables | Categories | Weighted frequencies and percentage | Weighted barriers to health care access (%) |
|---|---|---|---|
| Age of respondents | 15–24 | 66,532 (36.66) | 62.59 |
| | 25–34 | 59, 294(32.67) | 64.47 |
| | 35 and above | 55, 646(30.66) | 66.15 |
| Religion | Catholic | 33, 875(18.67) | 57.26 |
| | Protestant | 34, 652(19.10) | 51.92 |
| | Muslim | 44,345(24.44) | 54.99 |
| | Animist | 7,650(4.22) | 57.76 |
| | No religion | 2,759(1.52) | 53.42 |
| | Others[a] | 58,191(32.07) | 80.62 |
| Household wealth index | Poorest | 32,142(17.71) | 78.76 |
| | Poorer | 34,415(18.96) | 73.20 |
| | Middle | 35,074(19.33) | 68.01 |
| | Richer | 38,379(21.15) | 59.91 |
| | Richest | 41,463(22.85) | 46.44 |
| Educational status of the respondent | No education | 73, 756(40.64) | 76.96 |
| | Primary | 39,892(21.98) | 66.29 |
| | Secondary & Higher | 67,824 (37.37) | 48.27 |
| Covered by health insurance | Yes | 2,908(2.11) | 32.19 |
| | No | 134,751(97.89) | 66.48 |
| Parity | None | 45,028 (24.81) | 57.38 |
| | Multiparity | 75,184(41.43) | 62.19 |
| | Grandmultipara | 61,260 (33.76) | 64.31 |

[a] = Traditional, Orthodox

**Table 3. Community level factors of barriers to health care access in extremely high and very high maternal mortality countries (181, 472).**

| Variables | Categories | Weighted Frequencies and percentage | Weighted barriers to health care access |
|---|---|---|---|
| Residence | Rural | 112,568(62.03) | 72.59 |
| | Urban | 68,904(37.97) | 51.23 |
| Community level education | High | 95,942(52.87) | 58.39 |
| | Low | 85,530(47.13) | 70.46 |
| Community level media exposure | High | 102,412 (56.43) | 60.60 |
| | Low | 79,060 (43.57) | 68.94 |
| Community level poverty | High | 84,913(46.79) | 69.94 |
| | Low | 96,559(53.21) | 59.34 |
| Countries | Chad | 17,719 (9.76) | 84.34 |
| | Nigeria | 41,821 (23.05) | 51.52 |
| | Liberia | 8,065(4.44) | 44.65 |
| | Afghanistan | 29,461 (16.23) | 88.82 |
| | Lesotho | 6,621(3.65) | 41.77 |
| | Guinea | 10,874 (5.99) | 68.10 |
| | Democratic Republic of the Congo | 18,827 (10.37) | 76.03 |
| | Kenya | 32,156 (17.72) | 52.40 |
| | Benin | 15,928(8.78) | 60.37 |

(-2LLR). Model III was found to have the lowest deviance (148,967.536) that implies the best fitting model (Table 4).

## Factors associated with barriers to healthcare access among reproductive age women

In the final model, after adjusted for the individual and community level factors, age, educational level of the women, wealth index, media exposure, parity, sex of household head, type of place of health insurance, residence and country were the associated factors with barriers to healthcare access.

Accordingly, the odds of barriers to healthcare access was 1.36 times (95% CI: 1.31, 1.41) and 1.14 times (95% CI: 1.11, 1.18) higher among 15–24 and 25–34 years reproductive age women as compared to those women aged 35 and above years, respectively.

Those women who did not attend formal education were 2.12 times (95% CI: 2.05, 2.20) higher likelihoods of barriers to healthcare access as compared to those women who had attended secondary and higher education. Similarly, those women who had attended primary education had 1.41 times (95% CI: 1.36, 1.47) higher likelihoods of barriers to healthcare access.

The odds of barriers to healthcare access was 1.44 (95% CI: 1.38, 1.50), and 1.21 (95% CI: 1.17, 1.26) times higher among poor and middle wealth category households as compared to those women from the rich categories of households, respectively.

Those mothers who had no media exposure had 1.12 times (95% CI: 1.07, 1.14) higher likelihoods of barriers to healthcare access as compared with their counterparts.

The odds of barriers to healthcare access was 1.37 times higher among multiparous as compared with nulliparous (95% CI: 1.31, 1.44).

The odds of barriers to healthcare access was 2.06 times higher among women whose healthcare was not covered by health insurance as compared with those women whose healthcare service covered by health insurance (95% CI: 1.89, 2.25).

**Table 4. Multivariable analyses for factors affecting barriers to healthcare access among reproductive age women (181, 472).**

| Variables | Categories | Null model | Model 2 | Model 3 | Model 4 |
|---|---|---|---|---|---|
| | | | AOR (95% CI) | AOR (95% CI) | AOR (95% CI) |
| Age | 15–24 | | 1.39(1.34, 1.45) | | 1.36(1.31, 1.41)* |
| | 25–34 | | 1.15(1.11, 1.19) | | 1.14(1.11, 1.18)* |
| | ≥35 | | 1 | | 1 |
| Respondent education status | No education | | 2.43(2.35, 2.51) | | 2.12(2.05, 2.20)** |
| | Primary | | 1.55(1.49, 1.61) | | 1.41(1.36, 1.47)* |
| | Secondary & Higher | | 1 | | 1 |
| Current marital status | Married | | 1 | | 1 |
| | Unmarried | | 0.98(0.95, 1.02) | | 0.99(0.96, 1.03) |
| Wealth index | Poor | | 1.76(1.70, 1.82) | | 1.44(1.38, 1.50)* |
| | Middle | | 1.40(1.35, 1.45) | | 1.21(1.17, 1.26)** |
| | Rich | | 1 | | 1 |
| Media exposure | No | | 1.10(1.06, 1.13) | | 1.12(1.07, 1.14)* |
| | Yes | | 1 | | 1 |
| Parity | Nulliparity | | 1 | | 1 |
| | Multipara | | 1.09(1.05, 1.13) | | 1.05(0.98, 1.08) |
| | Grand multipara | | 1.40(1.34, 1.48) | | 1.37(1.31, 1.44)* |
| Covered by health insurance | No | | 2.12(1.94, 2.31) | | 2.06(1.89, 2.25)* |
| | Yes | | 1 | | 1 |
| Sex of household head | Male | | 1.11(1.07, 1.15) | | 1.09(0.98, 1.13) |
| | Female | | 1 | | 1 |
| Residence | Rural | | | 2.61(2.53, 2.68) | 1.73(1.66, 1.79)** |
| | Urban | | | 1 | 1 |
| Community level media exposure | Low | | | 1.06(0.95, 1.18) | 0.90(0.79, 1.02) |
| | High | | | 1 | 1 |
| Community level education | Low | | | 1.38(1.25, 1.52) | 1.09(0.97, 1.23) |
| | High | | | 1 | 1 |
| Community level wealth index | Low | | | 0.866(0.79, 0.95) | 1.02(0.91, 1.15) |
| | High | | | 1 | 1 |
| Countries | Extremely high | | | 0.68(0.66, 0.69) | 0.58(0.56, 0.59)*** |
| | Very high | | | 1 | 1 |
| **Random effect** | | | | | |
| **Variance** | | 0.778 | 0.692 | 0.763 | 0.584 |
| **ICC** | | 0.188 | 0.174 | 0.191 | 0.151 |
| **MOR** | | 2.31 | 2.20 | 2.29 | 2.06 |
| **PCV** | | Ref | 11.05 | 1.93 | 24.94 |
| **Model comparisson** | | | | | |
| **Loglikelihood (LLR)** | | -93899.322 | -90972.101 | -76580.884 | -74483.768 |
| **Deviance** | | 187,798.644 | 181,944.202 | 153,161.768 | 148,967.536 |

* = P-value < 0.05,

** = Pvalue < 0.01,

*** = Pvalue < 0.001;

ICC = Intra class corrolation cofficent, MOR = Median odds ratio, PCV = proportional change in variance. AOR = adjusted odds ratio; CI = confidence interval.

Rural resident women had 1.73 times higher odds of barriers to healthcare access as compared with their counterparts (95% CI: 1.66, 1.79).

The odds of barriers to healthcare access among extremely high maternal mortality countries was 42% times less as compared with those countries with very high maternal mortality (95% CI: 0.56, 0.59) (Table 4).

## Discussion

The overall, magnitude of barriers to health care access in countries with extremely high and very high maternal mortality was 64.31% (95% CI: 64.07, 64.55). This implies that only three in ten women had no barriers to healthcare access. The finding is inline with a study conducted in South Africa 64.5% [25]. However, studies conducted in sub Saharan Africa 61.5% [26] and in Ethiopia 61.3% [27] are lower than our findings. The possible reason for the higher magnitude of barriers to healthcare access might be due to study setting differences. The current study was conducted in a purposely selected high maternal mortality countries in contrast to the whole sub Saharan Africa. The finding implies there is a need to work in the improvement of health care access in countries with extremely high and very high maternal mortality.

The odds of barriers to healthcare access was higher among young aged women as compared with older (35 and above year) women. This is opposite with a study conducted in Ethiopia [27]. This might be because of lack of autonomy to healthcare utilization where families may take the role of decision to use healthcare for their children. As women age increases being independence to visit health facility may be increased.

Those women who did not attend formal education were reported higher odds of barriers to health care access. The finding is similar with other studies conducted in Ethiopia [27], Benin [28], South Africa [14], and East Africa [29]. The higher probability of barriers among non-educated might be due to lack of knowledge and awareness about benefit of visiting health facilities [28]. It is also obvious that education is the key factor for different opportunities such as employment, and economic growth that may in turn increases the accessibility of healthcare services [28].

Those reproductive age women who were from poor and middle households were reported higher likelihoods of barriers to healthcare access as compared to those reproductive age women who come from rich households. The finding was in line with studies conducted in Ethiopia [27, 30], sub-Sahara Africa [10], and Tanzania [30, 31]. The higher probability of barriers to healthcare access among poor wealth class might be accounted due to shortage of money or financial factors that determine the accessibility of healthcare services for a given households [14, 30].

The finding indicated that the likelihoods of barriers to healthcare access was higher among reproductive age women who had no media exposure as compared to their counterparts. The finding is similar with studies conducted in Pakistan [32], Nigeria [33], and Ethiopia [34]. This finding may be attributed to the fact that women without media exposure may not be aware of how to overcome access challenges that reduce their participation in reproductive health decisions [34].

The odds of barriers to healthcare access was higher among multipara women as compared with nulliparous. The finding is similar with a study conducted in Tanzania [35]. The lower barrier among primipara might be due to the risk of complications during pregnancy and delivery, which in turn leads to a higher consumption of maternal healthcare than multiparous women [36, 37]. It is possible that multiparous women have higher barriers due to undesirable experiences they may have had during their previous pregnancies and deliveries, including poor attitudes from health professionals, long waiting times, high service costs [38, 39].

Health insurance reinforces the utilization of healthcare services in a given population. Evidences indicated that insured persons are more likely to access and use healthcare services [40, 41]. In a similar manner the odds of barriers to healthcare access was higher among women whose healthcare was not covered by health insurance as compared with those women whose healthcare service covered by health insurance. This implies that policy makers should promote health insurance to minimize financial costs imposed to an individual.

Rural resident women had higher likelihoods of barriers to healthcare access as compared with their counterparts. Similarly, studies in sub Saharan Africa [42], and East Africa [29] revealed that rural resident women had higher likelihoods of barriers to healthcare access as compared to urban resident women. The possible reason for this might be physical accessibility of health infrastructures [30], limited education, and high transportation costs for rural resided population [29, 42]. Therefore, each respective country governments are expected to give attention for accessing healthcare services for rural resided mothers.

The odds of barriers to healthcare access was 42% times less as compared with those countries with very high maternal mortality. A possible reason may be the government's attention to countries with extremely high maternal mortality rates. Additionally, non-governmental partners may focus their attention on countries with extremely high maternal mortality rates in order to address healthcare access issues [43, 44].

## Strengths and limitations of the study

We used the recent nationally representative large sample size with a standardized and validated data collection tool. Hence, the finding can be generalized to extremely high and very high maternal mortality countries. The study tried to consider multi-level model to account for the clustering effect of DHS data and assessed basic individual and community level factors of barriers to healthcare access. However, the limitations of a cross-sectional study cannot be ruled out. Since the respondents were asked about the five-year event preceding the survey, the presence of recall and social desirability biases may also be high. There may be other indicators considered as barriers to healthcare. However, to construct the outcome variable, we used only the indicators available in the DHS data set. Therefore, we recommended the future researchers to conduct barriers to healthcare access by further adding other indicators such as culture and belief barriers to healthcare access with a mixed methods approach.

## Conclusion

More than six among ten reproductive age women had barriers to healthcare access in extremely high and very high maternal mortality countries. Lower age, no formal education, poor and middle wealth class, no media exposure, high parity, healthcare not covered by health insurance, rural residency and very high maternal mortality were significantly associated with barriers to healthcare access. Therefore, to improve access to health care services, extensive health education, expansion of health insurance and media exposure might be important interventions. Moreover, healthcare attentions to the poor wealth class women is another important intervention.

## Supporting information

**S1 File. Data used for the analysis of this study.**
(CSV)

## Acknowledgments

This study was made possible by permission from the DHS programs, for which we are grateful. We acknowledge Hiwot Tadesse Alemu for her initial overview of the manuscript.

## Author Contributions

**Conceptualization:** Wubshet Debebe Negash, Demiss Mulatu Geberu, Asebe Hagos, Amare Mesfin Workie.

**Data curation:** Wubshet Debebe Negash, Asebe Hagos, Kaleab Mesfin Abera, Abel Endawkie, Mihret Getnet.

**Investigation:** Wubshet Debebe Negash.

**Methodology:** Nigusu Worku.

**Software:** Wubshet Debebe Negash, Melak Jejaw, Misganaw Guadie Tiruneh.

**Supervision:** Demiss Mulatu Geberu, Kaleb Assegid Demissie.

**Validation:** Adina Yeshambel Belay, Lakew Asmare, Demiss Mulatu Geberu, Misganaw Guadie Tiruneh, Lamrot Yohannes.

**Visualization:** Wubshet Debebe Negash, Adina Yeshambel Belay, Lakew Asmare, Melak Jejaw, Kaleb Assegid Demissie, Misganaw Guadie Tiruneh, Kaleab Mesfin Abera, Yawkal Tsega, Nigusu Worku, Lamrot Yohannes, Mihret Getnet.

**Writing – original draft:** Wubshet Debebe Negash, Asebe Hagos, Abel Endawkie.

**Writing – review & editing:** Wubshet Debebe Negash, Lakew Asmare, Kaleb Assegid Demissie, Misganaw Guadie Tiruneh, Yawkal Tsega, Abel Endawkie, Nigusu Worku, Amare Mesfin Workie, Lamrot Yohannes, Mihret Getnet.

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
