## [Decision Letter · Decision Letter 0]

1 May 2024

PONE-D-24-02171More than six in ten mothers had barriers in healthcare access in extremely high and very high maternal mortality countries: Multilevel mixed effect analysisPLOS ONE

Dear Dr. Wubshet Debebe Negash,

Thank you for submitting your manuscript to PLOS ONE. After careful consideration, we feel that it has merit but does not fully meet PLOS ONE’s publication criteria as it currently stands. Therefore, we invite you to submit a revised version of the manuscript that addresses the points raised during the review process.

We look forward to receiving your revised manuscript.

Kind regards,

Gilbert Abotisem Abiiro, PhD

Academic Editor

PLOS ONE

Journal Requirements:

2. "We noticed you have some minor occurrence of overlapping text with the following previous publication(s), which needs to be addressed:

https://doi.org/10.1186/s12889-023-15952-w

In your revision ensure you cite all your sources (including your own works), and

quote or rephrase any duplicated text outside the methods section. Further consideration is dependent on these concerns being addressed.

3. In the online submission form, you indicated that [The data used for this study will be available with a reasonable request from the corresponding author.]. 

Additional Editor Comments:

The reviewer has raised a very important concern about the title of the manuscript. Could you please address the reviewer's concern?

Reviewers' comments:

Reviewer's Responses to Questions

**Comments to the Author**

1. Is the manuscript technically sound, and do the data support the conclusions?

Reviewer #1: Yes

2. Has the statistical analysis been performed appropriately and rigorously? 

Reviewer #1: Yes

3. Have the authors made all data underlying the findings in their manuscript fully available?

Reviewer #1: Yes

4. Is the manuscript presented in an intelligible fashion and written in standard English?

Reviewer #1: Yes

5. Review Comments to the Author

Reviewer #1: The study is interesting and reveals the barriers to healthcare access among reproductive age women.

It is not entirely clear to me why the author decided to use the title More than six in ten mothers had barriers in healthcare access in extremely high andvery high maternal mortality countries and not what was mentined in the abstract as the aim of the study (assess barriers to healthcare access among reproductive age women in extremely high and very high maternal mortality countries) please clarify.

6. PLOS authors have the option to publish the peer review history of their article (what does this mean?). If published, this will include your full peer review and any attached files.

Reviewer #1: **Yes: **EVERLINE DELYLAH ONDIEKI

---

## [Author Response · Author response to Decision Letter 0]

7 May 2024

Authors’ response to editor and reviewer comments 

We are very grateful to both the editor and reviewers for your comments and suggestions. All the concerns raised so far will have an undeniable impact on improving the quality and readability of our scholarly work. Appreciating your effort and valuable comments, we have provided possible reflections and amended the raised concerns and questions. Kindly find our reflections here.

We hope you will consider the revised manuscript acceptable for publication in PLOSE ONE research journal.

S.no Editor comments Authors’ responses

1 Please ensure that your manuscript meets PLOS ONE's style requirements, including those for file naming. The PLOS ONE style templates can be found at 

Dear Editor thank you for your comments. We revised the manuscript to meet PLOSE ONE’s style. 

2 We noticed you have some minor occurrence of overlapping text with the following previous publication(s), which needs to be addressed:

https://doi.org/10.1186/s12889-023-15952-w

In your revision ensure you cite all your sources (including your own works), and

quote or rephrase any duplicated text outside the methods section. Further consideration is dependent on these concerns being addressed. We tried to identify the minor overlaps and corrected it. 

3 In the online submission form, you indicated that [The data used for this study will be available with a reasonable request from the corresponding author.]. 

All PLOS journals now require all data underlying the findings described in their manuscript to be freely available to other researchers, either 1. In a public repository 2. Within the manuscript itself, or 3. Uploaded as supplementary information This policy applies to all data except where public deposition would breach compliance with the protocol approved by your research ethics board. If your data cannot be made publicly available for ethical or legal reasons (e.g., public availability would compromise patient privacy), please explain your reasons on resubmission and your exemption request will be escalated for approval. Thank you for your comments. We uploaded the data set as supplementary file 1. Kindly see on page 22 line 310 or page 25 line 442.

4 Your ethics statement should only appear in the Methods section of your manuscript. If your ethics statement is written in any section besides the Methods, please move it to the Methods section and delete it from any other section. Please ensure that your ethics statement is included in your manuscript, as the ethics statement entered into the online submission form will not be published alongside your manuscript. We accepted and putted the ethics statement in the methods section. Kindly see on page 10 line 155-165.

5 Please include captions for your Supporting Information files at the end of your manuscript, and update any in-text citations to match accordingly. Please see our Supporting Information guidelines for more information: http://journals.plos.org/plosone/s/supporting-information. 

Thank you for your recommendation. We include captions for supporting information at the end of the references. Kindly see on page 25 line 442.

6 Please review your reference list to ensure that it is complete and correct. If you have cited papers that have been retracted, please include the rationale for doing so in the manuscript text, or remove these references and replace them with relevant current references. Any changes to the reference list should be mentioned in the rebuttal letter that accompanies your revised manuscript. If you need to cite a retracted article, indicate the article’s retracted status in the References list and also include a citation and full reference for the retraction notice. We reviewed our references lists and ensured that they all are complete and correct.

7 Additional Editor Comments:

The reviewer has raised a very important concern about the title of the manuscript. Could you please address the reviewer's concern? Thank you. We revised the reviewer comments and corrected it accordingly. Kindly see on page 1 line 1-3.

 Reviewer #1: 

1 The study is interesting and reveals the barriers to healthcare access among reproductive age women.

It is not entirely clear to me why the author decided to use the title More than six in ten mothers had barriers in healthcare access in extremely high andvery high maternal mortality countries and not what was mentined in the abstract as the aim of the study (assess barriers to healthcare access among reproductive age women in extremely high and very high maternal mortality countries) please clarify. Dear Reviewer thank you for your important observations. We wrote the declarative form of the title. Because it is recommended that the title to be informative and more concise by findings. We did it for clarity and brevity. Our goal was to make the title more concise and conclusive so readers could identify the main message more easily. However, we accepted your comments and corrected the title as “Barriers to healthcare access among reproductive age women in extremely high and very high maternal mortality countries: Multilevel mixed effect analysis” Kindly see on page 1 line 1-3.

---

## [Editor Report · Decision Letter 1]

22 May 2024

Barriers to healthcare access among reproductive age women in extremely high and very high maternal mortality countries: Multilevel mixed effect analysis

PONE-D-24-02171R1

Dear Wubshet Debebe Negash ,

We’re pleased to inform you that your manuscript has been judged scientifically suitable for publication and will be formally accepted for publication once it meets all outstanding technical requirements.

Kind regards,

Gilbert Abotisem Abiiro, PhD

Academic Editor

PLOS ONE
---

## [Editor Report · Acceptance letter]

8 Jul 2024

PONE-D-24-02171R1 

PLOS ONE

Dear Dr. Negash, 

I'm pleased to inform you that your manuscript has been deemed suitable for publication in PLOS ONE. Congratulations! Your manuscript is now being handed over to our production team.

Kind regards, 

on behalf of

Dr. Gilbert Abotisem Abiiro 

Academic Editor

PLOS ONE